# Comparison of Chemical Composition and Safety Issues in Fish Roe Products: Application of Chemometrics to Chemical Data

**DOI:** 10.3390/foods9050540

**Published:** 2020-04-27

**Authors:** Mauro Vasconi, Erica Tirloni, Simone Stella, Chiara Coppola, Annalaura Lopez, Federica Bellagamba, Cristian Bernardi, Vittorio Maria Moretti

**Affiliations:** 1Department of Veterinary Medicine, Università degli Studi di Milano, Via dell’Università, 6, 26900 Lodi (LO), Italy; mauro.vasconi@unimi.it (M.V.); federica.bellagamba@unimi.it (F.B.); vittorio.moretti@unimi.it (V.M.M.); 2Department of Health, Animal Science and Food Safety, University of Milan, Via dell’Università, 6, 26900 Lodi (LO), Italy; erica.tirloni@unimi.it (E.T.); simone.stella@unimi.it (S.S.); chiara.coppola@unimi.it (C.C.); cristian.bernardi@unimi.it (C.B.)

**Keywords:** fish roes, fatty acids, organic acids, linear discriminant analysis, caviar

## Abstract

Processed fish roes are acquiring considerable importance in the modern food market, entering more and more often as an ingredient in food preparation and as caviar substitutes. In this study, we defined quality, traceability and safety issues related to processed fish roe products from different species. The results obtained allowed to distinguish eggs originated from different fish species and to discriminate between fish roes and caviar samples obtained from four different sturgeons species. We observed that roes showed a trend of grouping according to ecological and reproductive habits of fish species. We highlighted the differences between eggs originated by farmed and freshwater fish, enriched in n6 polyunsaturated fatty acids (PUFAs), and all the others, in which n3 PUFAs were prevalent. In addition, we evaluated processed fish roes under a food safety point of view, combining microbiological analysis with the determination of organic acids, used in some products as authorized preservatives. Microbiological characterization has proved a general good hygienic level for these products. Organic acids determination showed values in compliance with European Union (EU) regulations in almost of samples; in some cases, we found a mismatch between the organic acids detected and what was reported in labels. Processed fish roes could be considered a safe product that can provide to human nutrition a valuable content of essential fatty acids.

## 1. Introduction

Fish roe products has been historically present in many food cultures for a long time. The processing of mullet and tuna roes generates the Bottarga, a typical cured product of Mediterranean countries, while Ikura, Tarako and Tobiko are typical Japanese preparations that originate from salmon, pollock, and flying fish roes, respectively [1]. The most precious fish roe product is Caviar, obtained by light salting of roes extracted from sturgeons, separated from their connective tissue [2]. Caviar traditionally originated from the Russian and Persian empires and nowadays it is one of the most expensive luxury food products, appreciated all over the world. According to the Codex Alimentarius [3], only the product coming from the processing of roes obtained from fish of the Acipenseridae family can be named as Caviar. Fish roes from other fish species than sturgeon can be defined as caviar substitutes. They are sold at a lower price, but they refer to caviar with the aim of exploiting its high appeal among consumers. This reference is evidenced by the habit that some producers have when they confer the black color to products that naturally are differently colored, in order to resemble as much as possible to sturgeon roe [4]. In some cases, with traditional products like lumpfish roes, fish processed roes have a specific niche market that is not in competition with caviar market, but, on the contrary, they can lead new consumers to approach the consumption of fish eggs, starting with the cheapest products and then moving towards the product of excellence represented by caviar [5]. During last years, a great increase in the market of caviar substitutes occurred since (i) the availability of natural-sourced Caviar declined, due to the block of catch quotas granted in 2006 by the CITES (Convention on International Trade in Endangered Species) related to the steady decline of world sturgeon wild populations; (ii) the amount of Caviar produced by aquaculture was limited. In addition, the popularity of sushi led to the development of new products obtained by manufacturing roes from other fish species and to the expansion of their market. The estimated global market of processed fish roes covers 60,000 tons, while real caviar production does not reach 500 tons [4,6]. In 2000, European Union (EU) countries imported 5000 tons of frozen roes and 1000 tons of raw chilled roes from non-EU countries. At the same time, many EU countries (mainly in Denmark, Spain, and United Kingdom) also imported 7500 tons of prepared fish roe, valued at 119 M€ [7].

Egg processing techniques are different depending on the features of the raw material. The separation of eggs from the connective tissue could be achieved differently, according to egg dimensions. As an evidence, salmon and trout eggs are obtained by the manual separation, usually performed by a stainless steel grid [8], while smaller fish roes, such as capelin, herring, and pollock roes, are automatically processed in drum filters separators. Enzyme separation, using collagenase and pepsine, has also been studied [9,10]. Eggs are then salted, generally by a saturated salt solution, then maintained in barrels at refrigerated temperatures in order to decrease the water activity, thus allowing their shelf stability. Other additives or ingredients, like sugar, could be added in these processing phases [2]. After a suitable maturation period, eggs are packaged in glass or aluminum cans for the retail. In addition, eggs may be pasteurized at mild temperatures, in order to avoid the denaturation of proteins, as happens with lumpfish roes [11]. All fish roes are generally ready-to-eat products and they are sold under refrigerated conditions. Nevertheless, organic acids, like citric acid or benzoic acid, are usually added in fish roe as antimicrobials [2], although no studies aimed to investigate their presence are available in literature [12].

Since usually fatty acid analysis performed by mean of analytical chemistry leads to the yield of large amount of data, chemometrics methods appear to be necessary in order to investigate the structure of large datasets, especially when samples have different origins and can be divided into groups. This is the case of Principal Component Analysis (PCA), where the significant dimensionality of data matrices is strongly reduced, allowing the analyst to more easily understand the structure of chemical data, still retaining a good amount of the original information [13]. Moreover, qualitative chemometrics methods can be also very useful to create a classification model and to assign samples included in a dataset to a determinate group. In this scenario, multivariate methods appear to be very powerful, usually leading to a good fit of analytical data to a classification model. Particularly, the aim of Linear Discriminant Analysis (LDA) is to obtain an optimal separation among existing groups and, further, to maximize the prediction power in order to classify such samples whose membership is not known, on the basis of the distance of each observation from each group mean [13]. However, when the number of variables in the matrix is higher than the number of samples analysed, as often occurs with fatty acid analysis, it appears very advantageous to employ a PCA/LDA combined approach, in which firstly the dimensionality of raw data is reduced and then the discriminant model is built to a subset of the original variables [13].

Despite the importance of fish roe products in international market, there is a little technical information available about their chemical composition, food safety, and quality attributes. The aim of the present work was to achieve a deeper knowledge of the chemical composition, the fatty acid profile, and the organic acids content of several fish roe products collected on the Italian market. Moreover, we aimed to verify if a discrimination was achievable between caviar and fish roes by mean of a chemometric approach. Finally, microbiological properties of the products and the assessment of the conformity with the label information have been investigated and discussed.

## 2. Materials and Methods

### 2.1. Sampling

A total of 38 fish roe samples were purchased during the spring of 2016 (Table 1).

The survey involved 12 samples of Chum salmon (*Oncorhynchus keta*) eggs coming from 4 different producers; 11 samples of lumpfish (*Cyclopterus lumpus*) roe purchased from 2 different producers; 3 samples of rainbow trout (*Oncorhynchus mykiss*) roe; 3 samples of pike (*Esox lucius*) roe; 3 samples of cod (*Gadus morhua*) roe, 3 samples of Alaska pollock (*Theragra chalcogramma*) roe, and 3 of capelin (*Mallotus villosus*) roe, one black colored, one orange colored, and one green colored. All samples were packaged in glass jars or metallic cans and sold refrigerated. Samples were transported to laboratory under ice in polystyrene boxes, and then they were opened for the microbiological analyses. The remaining portion of samples was stored at −20 °C until chemical analyses.

### 2.2. Chemical Analyses

All chemicals and reagents were of analytical grade and were purchased from Sigma-Aldrich (Milan, Italy).

Before chemical analyses, three groups of eggs coming from each jar or can were weighed. Individual average weight of egg was calculated dividing the total weight with the number of eggs weighed.

The pH was measured using five grams of roes, blended with 20 mL of distillate water with an Ultraturrax for 30 s and pH was registered by a digital pH-meter (Amel Instruments, Milan, I) standardized at pH 4 and 7. Proximate composition and NaCl content were determined using standard methods [14].

The extraction and determination of total lipids were performed according to the Folch [15] method with chloroform: methanol mixture (2:1, *v/v*), using 0.7–1.0 g of sample, depending on its lipid content. The preparation of fatty acid methyl esters was performed according to Christie [16]. The sample was dissolved in 1 mL hexane and 1 µL sample was injected into the gas-chromatograph. Fatty acid analysis was carried out on an Agilent gas-chromatograph (Model 6890 Series GC) fitted with an automatic sampler (Model 7683) and FID detector (Agilent Technologies, Santa Clara, CA, USA) using helium as carrier gas. A HP-Innowax fused silica capillary column (30 m × 0.25 mm I.D., 0.25 µm film thickness, Agilent Technologies, Santa Clara, CA, USA) was used to separate fatty acid methyl esters. The oven temperature program for separation was from 100 to 180 °C at 3 °C min^−1^, then from 180 to 250 °C at 2.5 °C min^−1^ and held for 10 min. Fatty acids were identified relative to known external standards (Supelco 37 FAME Mix, code CRM47885, Marine source, code 47,033 and Menhaden fish oil code 47085-U, Supelco, Bellafonte PA, USA) and were expressed as percentage of total fatty acids.

### 2.3. Microbiological Analysis

For the microbiological determinations, 10 g of product were 10-fold diluted in chilled sterile diluent solution (0.85% NaCl and 0.1% peptone) and homogenized for 60 s in a Stomacher 400 (Seward Medical, London, UK). In some cases, due to the limited amount of material, pools of more than one sample were performed. Appropriate 10-fold dilutions of the homogenates were prepared in chilled saline solution. Total mesophilic viable count (TVC) was determined onto Plate Count Agar [17]. Lactic acid bacteria were enumerated on De Man Rogosa Sharpe (MRS) agar [18], *Enterobacteriaceae* were enumerated on Violet Red Bile Glucose Agar [19], yeasts and moulds were counted on Sabouraud Agar [20], coagulase positive staphylococci were counted on Baird Parker Agar [21], presumptive *Bacillus cereus* was enumerated onto PEMBA, *Enterococci* were enumerated onto Slanetz agar, (Merck, Darmstadt, Germany). Spores of reducing sulphite Clostridia were counted after pasteurization of the samples at 80 °C for 10 min onto Tryptose Sulfite Cycloserine, then incubated in anaerobiosis at 37 °C for 48 h. Detection and enumeration of *Listeria monocytogenes* were performed according to the method AFNOR [22]. All the culture media were purchased from Biogenetics (Ponte San Nicolò, Italy).

### 2.4. Determination of Organic Acids

All chemicals and reagents were of analytical grade and were purchased from Sigma-Aldrich (Milan, Italy).

The amounts of citric, lactic, and acetic acids were determined by Reverse phase-High Performance Liquid Chromatography (HPLC) analysis with ion-exchange chromatography and UV detection [23]. Briefly, 1 g of roe was added to 5.0 mL of water and vigorously shaken by a vortex for 20 s. After centrifugation, the supernatant was filtered through a 0.20 µm regenerated cellulose membrane filter. The HPLC system consisted of two pumps (510 by Waters S.p.A., Milano, Italy), an auto-sampler (717 plus by Waters S.p.A., Milano, Italy) and a UV-VIS detector (484 by Waters S.p.A., Milano, Italy) set at 210 nm. The separation was performed on a Rezex ROA column 300 mm × 7.8 mm, 8 µm (Phenomenex, Torrance, USA). The mobile phase (0.5 mL min^−1^ in isocratic mode) was 0.005 N sulphuric acid. External standards (acetic acid, code 1005706, Sigma Aldricht, lactic acid, code 252476, Sigma Aldricht and citric acid code 251275, Sigma Aldricht, Steinheim, Germany) were used for identification and quantification of acetic, citric, and lactic acid. The limit of detection (LOD: 0.076, 0.23, and 0.24 mM for acetic, citric, and lactic acid, respectively) and limit of quantification (LOQ: 0.39, 0.43, and 0.77 mM for acetic, lactic, and citric acid, respectively) were determined. LOD and LOQ of organic acids were calculated from the residual standard deviation of the regression line (SDrl) of the calibration curve and its slope (b) in accordance to the following equation: LOD = 3.3(SDrl/b) and LOQ = 10(SDrl/b) [24].

Concentration of benzoic and sorbic acids were determined by reverse phase HPLC method [25]. Briefly, 1 g of roes was homogenized with 10 mL of methanol:water (60:40 *v/v*) by an ultra-turrax and filled up to 20 mL with the extraction solvent. The mixture solution was placed in an ultrasonic bath for 30 min to complete the extraction, after centrifugation 1 mL of supernatant was filtered through a 0.45 μm regenerated cellulose (RC) membrane filter and fill up to 5 mL with the extraction solvent. The chromatographic analysis was carried out in an Alliance 2695 HPLC system equipped with a PDA 996 (Waters S.p.A., Milano, Italy) diode array detector. The separation was performed at a flow rate of 1 mL/min throughout a C18 Chromolith column, 100 × 4.6 mm, 5 μm, (Merck Millipore, Burlington, Massachusetts, USA). The mobile phase consisted of 5 mM ammonium acetate buffer pH 4.2 and methanol (70:30 *v/v*) and the eluent was monitored at 228 nm for benzoic acid and 260 nm for sorbic acid. External standards (sorbic acid code 47,845 Sigma Aldricht and benzoic acid, code 242,381 Sigma Aldricht, Steinheim, Germany) were used for identification and quantification of benzoic and sorbic acids. The limit of detection (LOD: 0.52, 0.50 mM for benzoic and sorbic acid, respectively) and limit of quantification (LOQ: 1.59 and 1.51 mM for benzoic and sorbic acid, respectively) were determined.

### 2.5. Statistical Analysis

Since data distribution was characterized by unequal variances within the groups (Levene test), statistical analysis was performed by mean of non-parametric tests (Welch test and Steel-Dwass post-Hoc test), declaring a significance when *p* < 0.05. A multivariate analysis was then performed by mean of Principal Component Analysis (PCA) on a 38 × 28 matrix, including fatty acid data, in order to reduce the dimensionality of data matrix and to detect similarities among samples and correlations among variables, according to Scano et al. [26]. Variables were selected when PC loadings score were > |0.5|. Then, a Linear Discriminant Analysis (LDA) was performed using the variables previously selected, in order to verify if they were satisfying in highlighting the differences among samples coming from different species. Since the distribution of the scores for Canonical-1 and Canonical-2 were confirmed for their normality and homoscedasticity, comparison among groups centroids scores were performed by mean of ANOVA and the Tukey-HSD test. Furthermore, in a successive step, thirty-seven caviar samples [27,28] were included in the dataset. The model was developed using roes from fish species analysed in this study as training set, including in the new matrix the variables selected in the previous steps, and caviar samples as validation set. Statistical analysis was performed using JMP Pro 14.0.0 (SAS Institute Inc., Cary, NC, USA).

## 3. Results and Discussion

### 3.1. Nutritional Quality and Authenticity of Fish Roes

Weights and proximate composition of eggs are presented in Table 2.

Egg size is influenced by the fish species of origin and by their reproductive behavior. Generally, fish with demersal spawning have bigger eggs if compared to species with planktonic eggs [29]. Biological features connected with the spawning site can also influence the chemical composition of roes that vary according to the maturation of fish from which they are obtained [30]. The protein content of processed fish roes analyzed in this study ranged from 8.1% to 29.6%, with salmon and trout eggs reaching the highest values (29.6% and 23.8%, respectively), lumpfish and capelin eggs showing the lowest values (10.8% and 8.1%, respectively), and pike, cod, and Alaska pollock eggs located in the middle (around 19%). According to their lipid content, samples could be divided in two groups. Salmon, trout, and pike roes recorded a lipid content > 12%, while all the other species showed a lipid content < 5%. An explanation for this phenomenon could be linked to the ecological habits of the various fish species. Fish that spawn in an environment that is poor of nourishment, like salmon and trout, produce eggs rich in lipid and protein, in order to guarantee an adequate nutrients supply to newborn generations for a prolonged period. Conversely, marine fish eggs (lumpfish, capelin, cod, Alaska pollock) contain minor reserves of nutrients, probably because the natural sites of spawning of these species are more able to easily provide food to newly hatched larvae. Ash content of roes ranged from 5.7%, recorded in capelin eggs, to 3.9%, recorded in pike eggs. The ash content is largely influenced by the amount of salt added to raw fish roes during their processing. Generally, our results agree with data previously reported by other authors, particularly regarding the species more commonly sold on the market, like salmon and lumpfish. Particularly, salmon roes analyzed in this work showed protein and lipid contents comparable to those reported by Bladsoe et al. in their review [2], whereas the lipid content was slightly lower than the one found in 2006 by Shirai et al. [1]. Proximate composition of lumpfish roes in our samples showed a lower protein content and a similar lipid content when compared with those reported by several authors in previous studies [11,31,32]. In salmon and lumpfish, the economic value of roes is higher than the one of meat [2], thus fish from these species are specifically harvested for roes production and their caught is planned when the roes reach the optimal maturation. Conversely, capelin, pollock, and cod roes are considered a valuable by-product of fish meat industry, so their composition could vary according to the harvesting season that is not strictly controlled as occurs for salmon and lumpfish. In capelin roes, Tocher and Sargent [30] reported a lipid content of 26.3%, corresponding to 7.89% on wet weight, which is superior to the value of 4.5% reported in the present paper. Regarding trout roes, we found a protein content slightly lower if compared with those reported by other authors [32,33,34,35,36] and a similar [34,35] or higher [2,32,33,34,37] lipid content. For pike roes, Bladsoe et al., [2] reported a protein content ranging from 14 to 27%, in agreement with the value of (19.4%) found in the present study; on the contrary they reported a lipid content between 1.5–2.4%, lower to the value (12.7%) of samples analyzed in the present survey. Proximate composition of cod roes characterized in our work was in agreement with data reported by Bladsoe et al. [2] for protein content, whereas lipid content agreed with data presented by Tocher and Sargent [30], recalculated as % of wet weight. Finally, Alaska pollock roes, used for the preparation of the typical Japanese fish preparation Tarako, showed a lipid content lower than the one reported by Shirai et al. [1] and a protein and lipid content similar to the one reported by Chiou et al. [38], recalculated on % wet weight.

Furthermore, in a previous study, we analyzed the chemical composition of caviar obtained from four different sturgeon species [27]. Caviar samples presented the mean moisture content of 54.71 ± 2.66 g/100 g and a protein content of 24.31 ± 1.51 g/100 g, while lipid content resulted 17.27 ± 2.81 g/100 g and ash content 3.70 ± 0.56 g/100 g. An interesting difference between caviar and its substitutes can be observed in the ash content that in caviar was around 3.7 and 3.9%, while in roes from other fish species was higher, up to 5.7%. Since the ash content is linked to the amount of salt and additives added to raw material during processing, a lower ash content demonstrated the tendency of using as less salt as possible in caviar production, related to the aim to obtain a high quality product.

Concerning protein content, salmon roes are the only analyzed roes that presented a higher amount of protein if compared with caviar, while trout roes showed a similar content.

Fatty acid (FA) composition in fish eggs is not directly related to the fatty acid composition of other fish tissues, as there is a certain selective fatty acid uptake in the ovarian tissues of female fish. As an evidence of this phenomenon, in 2017 Johnson et al. [39] tested the ability of Coho salmon (*Oncorhynchus kisutch*) of storing essential fatty acids, especially arachidonic acid (ARA) and docosahexaenoic acid (DHA), in the ovary during the secondary oocyte growth. The authors demonstrated that salmons incorporated the essential fatty acids supplied by the diets into the ovarian tissue rather than into muscle tissue. At the same time, it seems that fish are able to reallocate the essential FA stored in muscles to the ovary, if necessary, in order to satisfy the FA requirements of larvae, as reported by Zhu et al. [40].

All samples of fish roes analysed in this study (Table 3) showed a prevalence of polyunsaturated fatty acids (PUFA), reaching amounts up to 52.4%, on monounsaturated fatty acids (MUFA) and saturated fatty acids (SFA). PUFA are present in fish eggs in large amounts since they represent an optimal nutritional supply for the growth of the embryo and the larva [28]. Particularly, we recorded a relevant proportion of n3 series FA on n6 series FA in all samples. This proportion, commonly represented by the n3/n6 ratio, is considered fundamental by a nutritional point of view, either for the reproductive performance of fishes, and consequential larval growth, or for the nutritional quality of fish roes as favourable food product. The results obtained on the n3/n6 ratio in roes analysed in this study largely varied among the species considered. The highest value was recorded in lumpfish eggs (30.63), followed by capelin (17.35), salmon (15.14), cod (13.20), and Alaska Pollack (12.78) eggs, while the lowest values were found in pike (4.26) and trout (2.55) eggs.

The significant difference of the n3/n6 values among pike and trout roes and roes from the other species could be explained by the fact that trout and pike are freshwater species, living in an environment with lower n3 FA and higher n6 FA [41]. Moreover, trout roes originated from farmed fish; the use of plant derived oil in fish feed formulation involves an enrichment with n6 FA, mainly linolenic acid (LA, 18:2 n6), in farmed fish products [42]. As a consequence, trout roes showed the highest amount of LA, reaching 9.35% of total fatty acids. The trend found in n3 and n6 series FA in roes from these two species was similar to that found by Saliu et al. [43], who analyzed the fatty acid content of roes extracted from European catfish (*Silurus glanis)*, a freshwater predatory fish similar to pike for the habitat and the feeding habits. The authors found the prevalence of n3 FA on n6 FA but a lower n3/n6 ratio than in eggs of marine fish, recording a value of 3.4–3.8, comparable to the values we found in trout and pike roes.

n3 series FA in fish roes are mainly represented by eicosapentaenoic acid (EPA, 20:5 n3) and docosahexaenoic acid (DHA, 22:6 n3), particularly in marine species, reaching higher amounts than in fish flesh because of the higher phospholipids content of eggs [1]. Actually, long chain FA of the n3 series are conserved in roes at the expense of other FA, since they are valuable essential components of the biological membranes to be preserved during critical periods of larvae development [42] According to this, we found the highest content of both EPA and DHA in roes from lumpfish (18.93% and 26.81%), Alaska pollack (18.18% and 28.05%), capelin (16.07% and 19.27%), salmon (15.41% and 22.04%), and cod (17.80% and 27.14%), usually obtained after the caught of wild fish in natural stocks. At the same time, cod and Alaska pollack roes showed also the highest amount of arachidonic acid (ARA, 20:4 n6), another FA considered fundamental during starvation of marine fish larvae [44]. Capelin eggs showed high levels of the monounsaturated FAs palmitoleic acid (16:1) and gondoic acid (20:1 n-9). Nevertheless, the sum of eicosenoic fatty acids did not reach the levels found by Cyprian et al. [45], who analyzed the lipids from the whole capelin fish and found eicosanoic acids content near to 20% of total fatty acids. It can be suggested that these fatty acids do not have an essential role in larval metabolism and, for this reason, they are not stored in capelin roes. Pike eggs represented the only group in which MUFA were predominant (43.14%) on PUFA (36.02%). The prevalence of MUFA in pike eggs was mainly due to the higher content of 16:1 n7 (13.15%) and 18:1 n7 (7.67%) in samples if compared with eggs originating from other species. However, the most representative among MUFA in all analysed samples was oleic acid (OA, 18:1 n9). This fatty acid, together with the SFA palmitic acid (16:0), is known to represent the primary energy source for fish larvae in many species [46]. We found the highest amount of oleic acid in trout roes (27.11%). As stated before, trout is a farmed species and, thus, its diet is strongly enriched in vegetable oils that represent a source of many FA, less representative in natural aquatic trophic chains, such as oleic acid. As for LA, an increase in the ingestion of OA could have led to the higher stocking of this fatty acid in trout organs and eggs, mainly in the storage fraction represented by triacylglycerols [42]. 

Salmon eggs were rich of 18:3 n3 and DHA, even if they presented values of n3 fatty acids lower than other marine species as lumpfish, cod, and Alaska pollock. This aspect could be linked to the anadromous behavior of the species that partially lived in fresh waters during growing ages and during spawning season, but also to the feeding habits of the salmon species used to obtain roes. As a matter of fact, the Chum salmon *(O. keta*) feed mainly on copepods and euphausiids or gelatinous zooplankton, which have lower nutritional value than other prey [47].

In order to visualize the distribution of samples and eventual correlations between variables, a Principal Component Analysis was performed on the dataset including fatty acid data. The first two Principal Components (PC-1 and PC-2) introduced by mean of this multivariate technique explained the 60% of the total variance. The PCA loading plot (Figure 1) showed that 16 variables over the original 28 were related to a loading > |0.5|, thus highly influencing the variability of data and their distribution in the new bi-plot delimited by PC-1 and PC-2. In the loading plot, we can observe that the first PC, which explained the 37.4% of the variance, was mainly described by fatty acids typical of the marine habitat (EPA, DHA, n3/n6) in the positive direction and by fatty acids coming from the farm feeding system (oleic acid, linoleic acid, linolenic acid) in the negative direction. PC-1 significantly separated three groups: (i) pike and trout roes, inversely corelated with the n3/n6 ratio and positively correlated to fatty acids typical of freshwater environment or vegetable oils, such as oleic, linoleic and linolenic acid; (ii) lumpfish, Alaska Pollock and cod roes, from pure marine species, positively correlated with EPA and DHA and, consequently, with the n3/n6 ratio; (iii) salmon and capelin roes, in an intermediate position. The second PC, which explained the 24.7% of the total variance in the dataset, was mainly described by palmitic and palmitoleic acid in the positive direction and by stearic acid, oleic acid, 20:4 n3, and docosapentaenoic acid (DPA) in the negative direction. PC-2 was able to separate four groups: (i) capelin roes, positively correlated to palmitic and palmitoleic acid and saturated fatty acids in general; (ii) salmon and trout roes, positively correlated to 20:4 n3, DPA and stearic acid; (iii) lumpfish roes and (iv) pike, cod, and Pollock roes, in an intermediate position. However, the combination of both PC-1 and PC-2 in a multivariate system allowed distinguishing groups that would have not been recognizable if considering just one over the two direction, as demonstrated also by another study based on processed fish roes [48].

The variables selected by mean of PCA were then employed in the construction of a Linear Discriminant Analysis (LDA) model, performed in a new 38 × 16 matrix (38 samples and 16 fatty acids). The canonical plot obtained after the performance of the LDA is presented in Figure 2 whereas in Table 4, the Mahalanobis distances matrix is shown.

Sixteen variables were included in the model, in which Canonical 1 (Can-1) and Canonical 2 (Can-2) explained 83% of the total variance, so increasing the discriminant power if compared to the PCA previously performed. In order to investigate the influence of the variables on the construction of the discriminant model, their scoring coefficients were standardized and ordered by their weight on Can-1 and Can-2, respectively. In such LDA model, factors that most affected groups separation were 18:2 n6 and the n3/n6 ratio on Can-1; 18:1 n9, 22:5 n3, 22:6 n3 on Can-2; the total SFA, total MUFA, total PUFA, and total n3 FA, plus 20:5 n3, on both the canonicals. All roe samples coming from the different fish species analyzed in this study were clearly distinguishable by mean of the model built on the basis of fatty acid data, with the exception of cod and Alaska pollack roes that showed a strong overlapping among groups. As a matter of fact, the LDA model allowed a proper classification of all samples even if one Cod sample was assigned to the proper group with an uncertainty corresponding to 24%. The superimposition between Cod and Alaska Pollock samples is also observable in Table 4, where the Mahalanobis distance of Cod and Alaska Pollock centroids revealed that they were not significantly different (*p* > 0.05) when testing group means. These outcomes reflected the similarity observed by mean of univariate analysis on fatty acids composition. The two species, cod and Alaska Pollock, belong to the same Family, the *Gadidae*. They share the same environment represented by the cold waters of the northern hemisphere, the same reproductive habits and the same feed substrate; all these biological similarities resulted in a very close composition of their roes, not allowing our analyses to make a clear distinction between samples coming from their eggs. In the canonical bi-plot, samples belonging to groups cod, Alaskan pollock, lumpfish and capelin, pure marine species, were allocated in the same dial of the LDA chart related to n3 FA, particularly EPA and DHA, while pike and trout, freshwater and farmed species were in the opposite dial. Salmon showed a peculiar behavior, distancing itself from both marine and freshwater species. Salmon was the species that showed a higher value of n3 fatty acid, having the highest value of n3 fatty acid different from EPA and DHA, like eicosatetraenoic acid (ETA 20:4 n3) and docosapentaenoic acid (DPA 22:5 n3).

Finally, to ascertain whether the model would be able to distinguish among fish roes from these species, considered as caviar substitutes, and sturgeon caviar, we included in the dataset 37 samples of caviar previously analysed for fatty acid composition [27,28]. Then, we performed the same investigation, by mean of LDA, on the new matrix (75 × 16); the canonical plot obtained is showed in Figure 3.

We can observe that all roe samples coming from the various fish species analyzed in this study were clearly distinguishable from caviar. Caviar samples showed a more intra-group spread if compared to other fish roes samples. This phenomenon is due to origin of caviar samples, that were obtained from four different sturgeon species, which presented significative differences in their fatty acid composition [27]. Caviar showed a collocation in the plot close to trout roes samples. This phenomenon is linked to the origin of this samples; all of them came from farmed fish and their fatty acid signature is modified by the fatty acid of plan origin that characterize aquafeed, like linolenic acid (LA) and OA [28].

### 3.2. Food Safety of Fish Roes Products

According to the information found on labels placed on packaging showed (Table 1), salmon roes was the only product where no additive were added during processing. Pike, capelin, and lumpfish roes have been treated with thickeners, probably to give a more consistent texture to eggs. Capelin and lumpfish eggs were also treated to confer red, black, and green colors to these products, which naturally have a pale yellow color, being unattractive for consumers. Preservatives are generally added to fish roes in order to lower the pH, to prevent lipid oxidation, and to make eggs an unfavorable substrate for the growth of microorganisms. The decrease of the pH value contributes to maintaining the quality of fish roes during processing and storage [49]. During the extraction, fish roes should be considered sterile; however, they rarely remain sterile as microorganisms present on the surface of fish could be transferred to the roes, as well as the screening stage of eggs could contribute to contamination [2]. Thus, good manufacturing practices and a correct refrigeration maintenance are crucial to avoid bacterial replication during storage [49].

All the products analysed resulted safe as the presence of *L. monocytogenes* was never found. According to Reg. UE 2073/2005 [50], for ready-to-eat food with more than 5 days of shelf-life and able to support the *L. monocytogenes* growth, a tolerance level of 100 CFU/g is indicated during their shelf-life. Although fish roes have a short shelf-life, they are considered to support the growth of this potential pathogenic microorganism. As reported by Miya et al. [51], *L. monocytogenes* was found able to grow at 22 °C in 6 h; in the same study and differently from our findings, *L. monocytogenes* was found present with prevalence equal to 5.7% in salmon roes and 9.1% in cod roes, respectively. In a previous study, a prevalence of *L. monocytogenes* from 10 to 11.4% was found in fish roe products [52].

As shown in Figure 4 the total Viable Count (TVC) was generally very low, with 18 out of the 25 whole samples analysed (72%) showing values below the detection limit (2 Log CFU/g). Three out of the five samples of lumpfish roes analysed were countable with mean values equal to 2.26 Log CFU/g and all the capelin roes samples were countable with mean value equal to 3.14 Log CFU/g. The highest value of TVC detected was 5.00 and it was revealed in a sample of pike roes. The threshold limit of 6 Log CFU/g, frequently used for food products to designate the end of shelf-life of a fish product [53], was never exceeded. Moreover, values of total viable count above 7-8 Log CFU/g are often associated to sensory rejection. In agreement with our data, Oeleker et al. [54] evidenced a good microbiological situation with 54% of fish roes analysed in which total viable count below 2 Log CFU/g, while 84% loads below 4 Log CFU/g. The loads obtained in our study were in general lower if compared to those obtained by Altug and Bayrak, [55] for caviar from Russia and Iran, where TVC was from 3 to 6.41 Log CFU/g, and those obtained by Hilmelbloom and Crapo [56] in pink salmon ikura with loads from 3.48 to 6.48 Log CFU/g.

Lactic Acid Bacteria, *Enterobacteriaceae*, *B. cereus*, and *Enterococci*, resulted always below the detection limit (2 Log CFU/g). In all samples, Coagulase Positive *Staphylococci* resulted below the detection limit, except for a sample of lumpfish roes with a load of 3.88. Loads that are considered risky in terms of toxin production are generally set from 4 to 5 Log CFU/g, thus in our case, no particular concern resulted from their presence. Additionally, Oeleker et al. [54] evidenced a single sample containing Coagulase Positive Staphylococci where the load resulted just above 3 Log CFU/g.

Clostridia were always below the detection limit in all the samples analysed: as in fish roe, sterilization could not be applied due to the protein irreversible denaturation process, particular attention should be used to avoid the germination of possible spores present in the product, especially those belonging to *C. botulinum*. The maintenance of low refrigeration temperatures according with salt content are key factors in order to produce a safe product, able to inhibit *C. botulinum* germination. FDA suggests a Water Phase Salt (WPS) above 3.5%; in the products analysed, lumpfish roes, and capelin roes resulted to be characterized by a WPS below this limit (3.1 and 2.9%, respectively), thus the intrinsic characteristics of these two products could not guarantee the safety of the final product.

All pH values ranged from 5.12 to 6.28: at this pH range, benzoic acid may be effective against moulds and yeast in the range of 100–500 ppm [57]. It is fundamental to maintain a certain pH range in food products in order to inhibit the metabolic activity of spoilage microorganisms. Salmon and pike roes were found to be permissive for moulds replication since they had low benzoic acid and sorbic acid concentrations (<LOD equal to 52 ppm and 41 ppm, respectively) (Table 5). Moulds were present in all the samples of pike roes in very limited concentration (values from 2.00 to 2.60 Log CFU/g). At the same time, trout, cod, and Alaska Pollock roes were found to be permissive for mould replications even if high levels of sorbic acid were found in these samples. Even if in limited concentrations, we found moulds in all the samples of trout roes (all values equal to 2.00 Log CFU/g), in one sample of cod roes and in two samples of Alaska Pollack roes (loads equal to 2.00 Log CFU/g). Despite the fact that in a pH range between 4.0 and 6.0 sorbic acid is more effective than benzoic acid, due to their different pK (pK of sorbate = 4.80 vs. pK of benzoate = 4.20), sorbic acid has higher undissociated ratio in products with low acidity if compared to benzoic acid. For this reason, we isolated moulds and yeasts also in products characterized by the presence of both sorbic and benzoic acid but a low acidity, like capelin roes. In two samples of capelin roes, we found yeasts with loads corresponding to 2.00 and 2.30 Log CFU/g: high values of yeasts >5 Log CFU/g are recognized to cause organoleptic spoilage that in our samples was never detected. Sorbic and benzoic acids maximum level permitted in EU (2000 ppm) is applicable to both of them, with the final levels expressed as free acid [58]. Cod and pollack roes were characterized by sorbic acid concentrations that exceed the limit, whereas trout roes were found to have sorbic acid concentration close to the maximum. All the other products were in compliance to the EU regulation. Six out of eleven lumpfish products declared potassium sorbate (E202) in the list of ingredients, but sorbate salt was not detected in those samples. Cod roes did not declare benzoic acid (E211–E212) on the labels, however they all contained this acid, even if in low concentration.

Other organic acids such as citric, lactic, and acetic acid and relative salts are permitted in fish roes with the principle of *quantum satis*. These organic acids are usually used to lower the pH in food. The order of their effectiveness towards pathogens is acetic > lactic > citric. As was the case for sorbic and benzoic acid, some of the labels did not match our results. Citric acid was found in trout and pike roes (898 ppm and 2695 pmm, respectively) but it was not reported on the labels. Citric and lactic acid (E270) was found in all the samples, ranging from 232 ppm (pike) to 11,752 ppm (Alaska pollock), but none of the products declared lactic acid on the label. Acetic acid (E260) was found in salmon, trout, cod, and Alaska pollock, but even in this case the preservative was not declared. The presence of these undeclared organic acid is not a safety concern but an unfair commercial practice. In Alaska pollock, cod, and trout, the organic acids mixture seem very effective against microbial growth. No data about organic acids in fish roes are available in literature; however, these organic acids were used to build predictive model for *L. monocytogenes* in lightly preserved and ready-to-eat seafood [59].

## 4. Conclusions

This study was undertaken to provide the seafood industry and the consumers with information on nutritional and safety properties of fish roe preparations on the market, using a chemometric approach. Proximate composition showed that these products are extremely variable and that their composition depends on the biology of fish, their reproductive environment, and their food habits. Furthermore, this study clearly indicates that the combination of fatty acid composition along with a chemometric approach can be successfully applied to give more information on the original fish species of roes. Generally, fish roes are very rich in essential fatty acid, especially EPA and DHA, and they could be considered a good source of these substances in human diets. Moreover, all the product analyzed were considerable safe for human consumption, since the low microbial loads detected, probably related to the presence of organic acids added to the products during manufacturing processes, were associated with a very limited replication of the present microorganisms.

## Figures and Tables

**Figure 1 foods-09-00540-f001:**
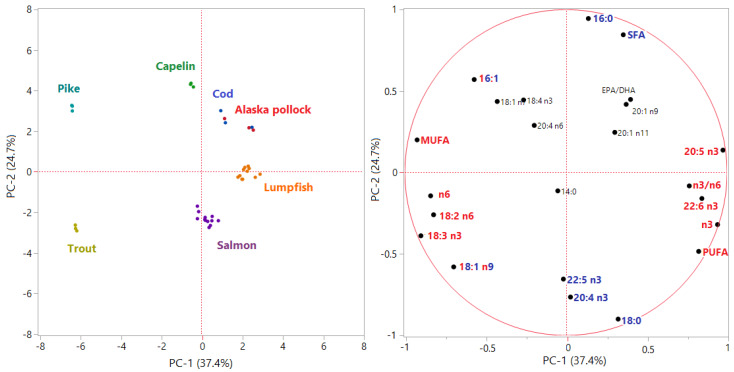
Principal Component Analysis score plot (left) and loading plot (right). In the loading plot, variables with loadings > |0.5| on PC-1 are in red, whereas variables with loadings > |0.5| on PC-2 are in blue.

**Figure 2 foods-09-00540-f002:**
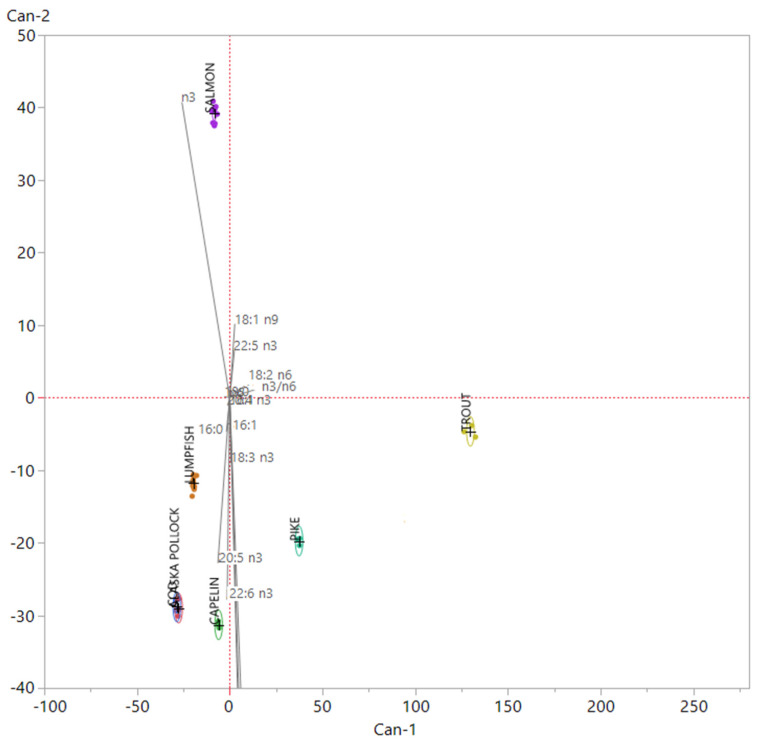
Linear Discriminant Analysis canonical plot. Each group is related with a 95% confident region density ellipse. In this graph, the scoring coefficients of the variables have been standardized and then multiplied by a scaling factor corresponding to 4.5, in order to better fit the canonical plot.

**Figure 3 foods-09-00540-f003:**
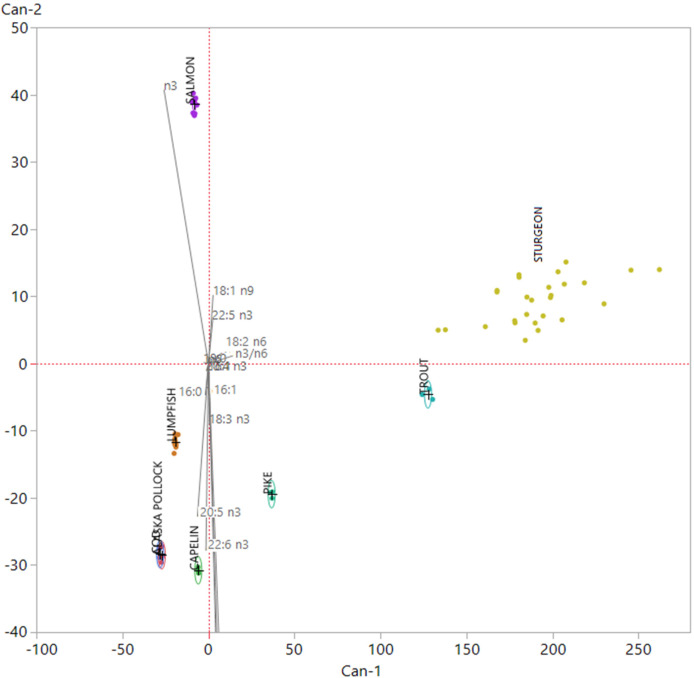
Canonical plot obtained after the employment of the linear discriminant analysis (LDA) model using fish roes from various species as training set and including 37 sturgeon caviar samples [27,28] as validation set.

**Figure 4 foods-09-00540-f004:**
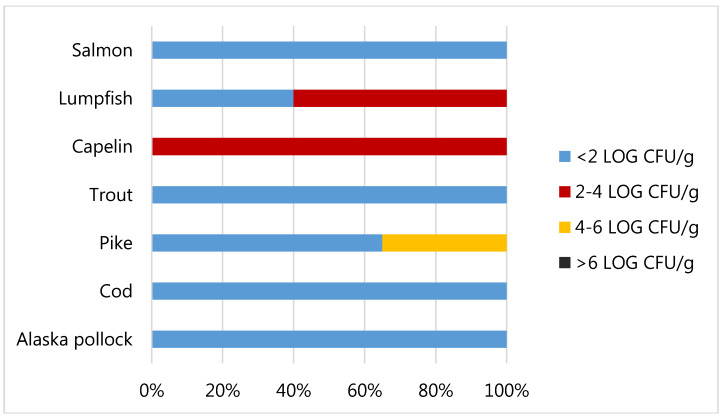
Total Viable Count of fish roe samples.

**Table 1 foods-09-00540-t001:** Fish roes samples collected in the study with the additives listed according to what was reported in their labels.

Species	Samples		Different Preparations					
	Total Number	Number Per Producer	NaCl *	Stabilizers	Preservatives	Antioxidants	Color Additives	Others
Salmon *Oncorhynchus keta*	12	3	2.5%	ND	ND	ND	ND	ND
		3	3.5%	ND	ND	ND	ND	ND
		3	2.5%	ND	ND	ND	ND	ND
		3	ND	ND	ND	ND	ND	ND
Lumpfish *Cyclopterus lumpus*	11	2	ND	E413	E211	E330	E120, E160c	Chartamus
		3	ND	E413	E211	E330	E150d, E151	
		3	ND	E412, E422	E202, E211	E331	E141, E150d, E163, E151	E621 (flavour enhancer)
Capelin *Mallosus villosus*	3	3	ND	E412, E422	E202, E212	E330	E120, E160c	E622 (flavour enhancer)
		1	ND		E202, E211	E330	E150d	Sugar
		1	ND	E413	E202, E211	E330	E160a, E120	Sugar, soy
Rainbow trout *Oncorynchus mykiss*	3	3	<6%	ND	E200	ND	ND	ND
Pike *Esox lucius*	3	3	ND	E415	ND	ND	ND	ND
Cod *Gadus morhua*	3	3	ND	ND	E200	ND	ND	ND
Alaska Pollock *Theragra chalcogramma*	3	3	ND	ND	E200	ND	ND	ND

* NaCl was present in all samples, table shows the amount as reported on the labels. Legend for the ingredients: ND: Not declared; E 412 Guar gum; E413: Tragacanth gum; E415 Xanthan gum; E422 Glycerol; E200: Sorbic acid; E202: Potassium sorbate: E211; Sodium benzoate: E212; Potassium benzoate; E330: Citric acid; E331: Sodium citrate; E102: Tartrazine; E120: Carminic acid; E133: brilliant blue FCF; E141: Chlorophylls and chlorophyllins, copper complexes; E150d: Sulfite Ammonia Caramel; E151: Brilliant Black BN; E160a: Carotenes; E160c: paprika oleoresin; E163: Anthocyanins; E621: Monosodium L-glutamate; E622: Monopotassium L-glutamate.

**Table 2 foods-09-00540-t002:** Weight (mg), proximate composition (g/100 g), and salt (g/100 g) content of fish roes. Data are expressed as mean ± standard deviation.

Fish Species	Salmon *n* = 12	Lumpfish *n* = 11	Capelin *n* = 3	Trout *n* = 3	Pike *n* = 3	Cod *n* = 3	Alaska Pollock *n* = 3
Weight mg	217.3 ± 31.33	6.0 ± 1.07	0.7 ± 0.09	54.7 ± 7.36	4.7 ± 0.50	0.2 ± 0.04	0.2 ± 0.02
Moisture g/100 g	53.5 ± 1.68	79.7 ± 0.92	81.7 ± 0.77	59.2 ± 0.03	64.0 ± 0.83	71.6 ± 0.55	73.5 ± 0.42
Protein g/100 g	29.6 ± 0.85	10.8 ± 0.40	8.1 ± 0.56	23.8 ± 0.07	19.4 ± 1.12	19.6 ± 0.27	19.2 ± 0.50
Lipid g/100 g	12.8 ± 0.85	4.3 ± 0.58	4.5 ± 0.23	12.5 ± 0.08	12.7 ± 0.57	3.2 ± 0.17	2.8 ± 0.10
Ash g/100 g	4.2 ± 0.79	5.2 ± 0.89	5.7 ± 0.02	4.6 ± 0.05	3.9 ± 0.04	5.5 ± 0.11	4.4 ± 0.05
NaCl g/100 g	2.7 ± 0.64	2.7 ± 0.64	2.5 ± 0.32	2.3 ± 0.06	3.3 ± 0.00	3.1 ± 0.06	2.8 ± 0.04

**Table 3 foods-09-00540-t003:** Fatty acid composition (g/100 g of total fatty acids) of fish roe from different species purchased in the Italian market. Data are expressed as mean ± standard deviation.

Fish Species	Salmon *n* = 12	Lumpfish *n* = 11	Capelin *n* = 3	Trout *n* = 3	Pike *n* = 3	Cod *n* = 3	Alaska Pollak *n* = 3
14:0	3.44 ± 0.51 ^b^	1.37 ± 0.11 ^c^	4.54 ± 0.01 ^a^	1.58 ± 0.04 ^c^	1.76 ± 0.03 ^c^	1.47 ± 0.04 ^c^	1.68 ± 0.19 ^c^
15:0	0.54 ± 0.11^a^	0.27 ± 0.03 ^bc^	0.36 ± 0.00 ^bc^	0.19 ± 0.00 ^c^	0.27 ± 0.00 ^bc^	0.30 ± 0.01 ^bc^	0.38 ± 0.11 ^b^
16:0	10.63 ± 0.43 ^d^	14.19 ± 0.37 ^c^	17.07 ± 0.09 ^ab^	10.85 ± 0.20 ^d^	16.05 ± 0.12 ^b^	17.59 ± 0.81 ^a^	17.44 ± 0.56 ^ab^
16:1 n7	4.83 ± 0.53 ^c^	1.82 ± 0.25 ^e^	10.59 ± 0.10 ^b^	2.93 ± 0.10 ^d^	13.15 ± 0.30 ^a^	4.89 ± 0.21 ^c^	4.93 ± 0.14 ^c^
16:2 n4	0.14 ± 0.03 ^c^	0.02 ± 0.06 ^d^	0.36 ± 0.00 ^b^	0.13 ± 0.02 ^c^	0.70 ± 0.03 ^a^	nd	nd
17:0	0.51 ± 0.10 ^a^	0.41 ± 0.07 ^ab^	0.08 ± 0.13 ^d^	0.19 ± 0.00 ^bcd^	0.40 ± 0.01 ^abc^	0.25 ± 0.29 ^bcd^	0.14 ± 0.24 ^cd^
16:3 n4	0.85 ± 0.33 ^a^	0.02 ± 0.06 ^c^	nd	0.05 ± 0.04 ^bc^	0.53 ± 0.02 ^ab^	0.08 ± 0.13 ^bc^	nd
18:0	4.76 ± 0.23 ^a^	4.37 ± 0.19 ^b^	2.29 ± 0.01 ^b^	3.90 ± 0.06 ^c^	2.36 ± 0.04 ^de^	2.83 ± 0.18 ^d^	2.90 ± 0.28 ^d^
18:1 n9	18.25 ± 1.55 ^c^	17.07 ± 1.81 ^c^	11.21 ± 0.11 ^d^	27.11 ± 0.22 ^a^	21.65 ± 0.20 ^b^	12.50 ± 0.60 ^d^	12.82 ± 0.59 ^d^
18:1 n7	2.85 ± 0.40 ^b^	4.02 ± 0.45 ^b^	4.43 ± 0.04 ^ab^	3.28 ± 0.06 ^b^	7.67 ± 0.14 ^a^	4.25 ± 3.68 ^b^	2.09 ± 3.62 ^b^
18:2 n6	1.32 ± 0.11 ^d^	1.04 ± 0.21 ^e^	1.72 ± 0.01 ^c^	9.35 ± 0.19 ^a^	3.50 ± 0.04 ^b^	0.53 ± 0.03 ^f^	0.61 ± 0.04 ^f^
18:3 n3	1.22 ± 0.14 ^c^	0.39 ± 0.06 ^e^	0.78 ± 0.01 ^d^	3.53 ± 0.04 ^a^	2.20 ± 0.11 ^b^	nd	nd
18:4 n3	1.07 ± 0.11 ^c^	0.88 ± 0.26 ^cd^	3.08 ± 0.02 ^a^	0.66 ± 0.02 ^de^	1.73 ± 0.05 ^b^	0.46 ± 0.03 ^e^	0.47 ± 0.04 ^e^
18:4 n1	0.34 ± 0.10 ^a^	nd	nd	0.19 ± 0.00 ^b^	nd	nd	nd
20:1 n11	0.43 ± 0.15 ^c^	nd	nd	nd	nd	1.69 ± 0.03 ^b^	2.01 ± 0.23 ^a^
20:1 n9	0.85 ± 0.27 ^d^	4.06 ± 0.92 ^ab^	5.10 ± 0.08 ^a^	2.85 ± 0.06 ^c^	0.41 ± 0.02 ^d^	2.79 ± 0.05 ^c^	3.05 ± 0.17 ^bc^
20:1 n7	0.32 ± 0.03 ^b^	0.54 ± 0.20 ^a^	nd	0.03 ± 0.05 ^cd^	0.26 ± 0.01 ^bcd^	0.34 ± 0.02 ^abc^	0.12 ± 0.21 ^bcd^
20:2 n6	0.32 ± 0.05 ^a^	nd	nd	1.64 ± 0.03 ^a^	0.36 ± 0.00 ^a^	nd	0.16 ± 0.28 ^c^
20:3 n6	0.12 ± 0.05 ^c^	nd	nd	0.75 ± 0.02 ^a^	0.42 ± 0.00 ^b^	nd	nd
20:4 n6	1.42 ± 0.23 ^c^	0.64 ± 0.16 ^d^	0.68 ± 0.00 ^d^	1.40 ± 0.04 ^c^	2.34 ± 0.02 ^b^	3.07 ± 0.16 ^a^	3.07 ± 0.19 ^a^
20:4 n3	2.51 ± 0.19 ^a^	0.97 ± 0.15 ^b^	0.73 ± 0.00 ^bc^	0.94 ± 0.02 ^bc^	0.62 ± 0.01 ^c^	0.12 ± 0.22 ^d^	nd
20:5 n3	15.41 ± 0.53 ^b^	18.93 ± 0.77 ^a^	16.07 ± 0.07 ^b^	6.42 ± 0.02 ^d^	9.03 ± 0.03 ^c^	17.80 ± 1.07 ^a^	18.18 ± 0.56 ^a^
22:1 n9	0.19 ± 0.12 ^b^	0.49 ± 0.18 ^a^	nd	nd	nd	nd	nd
22:5 n3	5.56 ± 0.47 ^a^	1.70 ± 0.16 ^c^	1.64 ± 0.02 ^c^	2.02 ± 0.01 ^bc^	2.38 ± 0.01 ^b^	1.90 ± 0.15 ^bc^	1.90 ± 0.07 ^bc^
22:6 n3	22.04 ± 1.20 ^b^	26.81 ± 1.53 ^a^	19.27 ± 0.39 ^c^	20.01 ± 0.37 ^bc^	11.91 ± 0.29 ^c^	27.14 ± 1.21 ^a^	28.05 ± 1.09 ^a^
SFA	19.88 ± 0.41 ^e^	20.61 ± 0.63 ^d^	24.33 ± 0.18 ^a^	16.70 ± 0.22 ^f^	20.84 ± 0.17 ^cd^	22.44 ± 1.05 ^bc^	22.54 ± 1.24 ^b^
MUFA	27.72 ± 1.71 ^cd^	28.00 ± 1.78 ^d^	31.34 ± 0.23 ^c^	36.21 ± 0.29 ^b^	43.14 ± 0.24 ^a^	26.47 ± 2.83 ^d^	25.02 ± 2.82 ^d^
PUFA	52.40 ± 1.66 ^a^	51.39 ± 1.28 ^a^	44.33 ± 0.36 ^b^	47.09 ± 0.47 ^b^	36.02 ± 0.40 ^c^	51.09 ± 2.51 ^a^	52.44 ± 1.79 ^a^
n3	47.88 ± 1.43 ^a^	49.68 ± 1.06 ^a^	41.57 ± 0.37 ^b^	33.57 ± 0.34 ^c^	28.17 ± 0.41 ^d^	47.43 ± 2.44 ^a^	48.60 ± 1.71 ^a^
n6	3.19 ± 0.31 ^d^	1.67 ± 0.35 ^e^	2.40 ± 0.01 ^d^	13.15 ± 0.22 ^a^	6.62 ± 0.03 ^b^	3.59 ± 0.18 ^c^	3.84 ± 0.47 ^c^
n3/n6	15.14 ± 1.26 ^b^	30.63 ± 5.19 ^a^	17.35 ± 0.24 ^b^	2.55 ± 0.05 ^c^	4.26 ± 0.07 ^c^	13.20 ± 0.18 ^b^	12.78 ± 1.53 ^b^

a, b, c, d = Value within the same raw not sharing a common letter are significantly different (*p* < 0.05); SFA = saturated fatty acids; MUFA = monounsaturated fatty acids; PUFA = polyunsaturated fatty acids. nd = not detected.

**Table 4 foods-09-00540-t004:** Mahalanobis distances calculated as Euclidean distances among groups centroids in the canonical plot defined by Canonical-1 and Canonical-2 on the x- and y- axis, respectively.

	Salmon	Lumpfish	Capelin	Trout	Pike	Cod
Lumpfish	52.3					
Capelin	70.6	23.7				
Trout	144.9	149.4	138.3			
Pike	74.7	57.4	44.9	93.6		
Cod	71.0	19.3	22.7	160.1	66.6	
Alaska pollock	17.3	6.9	21.6	157.2	64.9	0.9 *

* Comparison among group means on Canonical-1 and Canonical-2 scores did not show any statistical difference between Cod and Alaska Pollock centroids (*p* > 0.05).

**Table 5 foods-09-00540-t005:** pH and organic acids (mg/kg) content of fish roes. Data are expressed as mean ± standard deviation.

Fish Species	Salmon *n* = 12	Lumpfish *n* = 11	Capelin *n* = 3	Trout *n* = 3	Pike *n* = 3	Cod *n* = 3	Alaska Pollock *n* = 3
pH	6.28 ± 0.08	5.77 ± 0.09	6.19 ± 0.06	5.95 ± 0.03	5.88 ± 0.05	5.96 ± 0.06	5.95 ± 0.03
Citric acid ppm	nd	nd	1356 ± 68	898 ± 75	2695 ± 192	<LOQ	<LOQ
Lactic acid ppm	676 ± 261	5343 ± 2472	1546 ± 172	10,812 ± 746	232 ± 7	11,072 ± 370	11,752 ± 2168
Acetic acid ppm	263 ± 92	nd	nd	713 ± 50	<LOD	875 ± 75	1177 ± 56
Sorbic acid ppm	nd	nd	607 ± 70	2021 ± 35	<LOD	2050 ± 118	2573 ± 491
Benzoic acid ppm	<LOD	858 ± 347	598 ± 120	<LOD	<LOD	285 ± 68	<LOD

LOD: Limits of Detection were 216 ppm for citric acid, 16 ppm for lactic acid, 47 ppm for acetic acid, 41 ppm for sorbic acid, and 52 ppm for benzoic acid. LOQ: Limits of Quantification were 722 ppm for citric acid, 52 ppm for lactic acid, 157 ppm for acetic acid, 151 ppm for sorbic acid, and 159 for benzoic acid.

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
