# Peer review of "Comparison of Chemical Composition and Safety Issues in Fish Roe Products: Application of Chemometrics to Chemical Data"

_foods, 2020, doi:10.3390/foods9050540_

Round 1
Reviewer 1 Report
Dear Authors,
this article is interesting and worth of publication, after a re-organization of the result presentation and statistical elaboration that are not clearly exposed even if Discriminant Analysis is appropriate in this case, with the normalization of data, as correctly proposed by authors.
The main weakness of this paper is the inclusion of caviar samples coming from previous research. These data must be eliminated by the main statistical elaboration that must be focused on your own samples and those caviar analyses could be eventually added in a successive figure. The aim of this interesting research was to analyse the fish eggs samples and the paper must be focused on this topic. Consequently, caviar comparison must be eliminated by the title.
Authors should not be discouraged by an eventual difficulty in the interpretation of statistical elaboration of their own samples (only caviar substitutes) as the reader will understand this eventual difficulty. In this paper, the sensation of the reader is that the authors included caviar samples in order to maximize intra-group differences and overcome possible difficulties in results interpretation. For this reason, caviar samples must be eventually considered only in a second step analysis.
There are some weaknesses in the manuscript, I included my comments directly in the paper.
Authors should include some more references, as I suggested in the article.
English must be improved.

Author Response
The authors are thankful to the reviewer for the kind comments. They were a precious help for the improvement of ours manuscript. Here a punctual reply to any comment.
Comment 1
this article is interesting and worth of publication, after a re-organization of the result presentation and statistical elaboration that are not clearly exposed even if Discriminant Analysis is appropriate in this case, with the normalization of data, as correctly proposed by authors.
Response
We thank the reviewer for appreciating the amount of work that led to the writing of this article. In agreement with reviewer suggestion, we improved the approach to the statistical analysis. In a first step, we developed a PCA in order to detect the variables that mostly contributed to the variance in our dataset. Consequently we included only the fish roes from the species under investigation in a LDA model. Finally, we introduced caviar samples to underline the differences between fish roes from several species and caviar . We hope that this new presentation of results could be more clear and satisfy the reviewer request.
Comment 2
The main weakness of this paper is the inclusion of caviar samples coming from previous research. These data must be eliminated by the main statistical elaboration that must be focused on your own samples and those caviar analyses could be eventually added in a successive figure. The aim of this interesting research was to analyse the fish eggs samples and the paper must be focused on this topic. Consequently, caviar comparison must be eliminated by the title.
Authors should not be discouraged by an eventual difficulty in the interpretation of statistical elaboration of their own samples (only caviar substitutes) as the reader will understand this eventual difficulty. In this paper, the sensation of the reader is that the authors included caviar samples in order to maximize intra-group differences and overcome possible difficulties in results interpretation. For this reason, caviar samples must be eventually considered only in a second step analysis.
Response
In a preliminary drafting of the manuscript we did not include caviar samples in our analysis, then, in order to underline the differences between fish roes and caviar, we included that samples in our analysis, finding that the separation of the seven fish species was not compromised by this addition. Caviar samples are so well separated and distinguished from other fish roes samples that its presence did not influence the behaviour of the remaining samples. However we accept with gratitude this suggestion and we performed the statistical analysis in two successive stages, the fist one considering fish roes alone underling the differences among species, and then we included caviar to show how caviar samples are deeply different from all the other species.
There are some weaknesses in the manuscript, I included my comments directly in the paper.
Response
All the comments and suggestion showed in attached file were satisfied and integrated into the new version of the manuscript
Authors should include some more references, as I suggested in the article.
Response
The references suggested has been added to the manuscript
English must be improved.
Response
All the document has been revised to increase the quality of English
Reviewer 2 Report
Why these fish species were selected? Also, please explain the importance of chemometrics in introduction.
L69: Fish roe is a term for unprocessed fish eggs, and I would like to suggest to change this to either fish caviar or processed roe.
L174: Usually, size is the most important factor compared to weight, also, it is very obvious that the largest eggs belong to salmon, and smallest ones belong to pelagic species. I believe this section is not informative and it is more descriptive.
Compositions and Table 1: The origin of these species are not clear. Since they are from different processors and possibly different origin, there will be different in quality and proximate.
L336: Please revise, it does not read well.
L349: This is the red line for any caviar processor and I dont expect any L. monocytogenes in any caviar product and in EU is 100 cfu/g.
L352: fish caviar, please do not use roe for processed eggs.
Author Response
The authors are thankful to the reviewer for the kind comments. They were a precious help for the improvement of ours manuscript. Here a punctual reply to any comment.
Comment 1
Why these fish species were selected? Also, please explain the importance of chemometrics in introduction.
Response
The fish species used in present study were the main fish present on the Italian market of fish roes product. Since salmon and lumpfish are the species with a larger diffusion, several producers of these preparations have been sampled. As we did not found any variation between processed eggs derived from the same species but sold by different producers, we did not discuss any manufacturer differences in our manuscript, but we analysed only the differences among fish species.
A paragraph about the importance of chemometrics has been added.
Comment 2
L69: Fish roe is a term for unprocessed fish eggs, and I would like to suggest to change this to either fish caviar or processed roe.
Response
Usually the term caviar is used for all fish egg products, but according to what established by the Codex Alimentarius the term caviar is reserved for processed eggs coming from sturgeons. For this reason in our article we used the term caviar only when we refer to “real” caviar, originated from un-ovulated sturgeons eggs. According to your suggestion we changed “roes” with “processed roes” when necessary.
Comment 3
L174: Usually, size is the most important factor compared to weight, also, it is very obvious that the largest eggs belong to salmon, and smallest ones belong to pelagic species. I believe this section is not informative and it is more descriptive.
Response
As suggested by the reviewer we delete the paragraph.
Comment 4
Compositions and Table 1: The origin of these species are not clear. Since they are from different processors and possibly different origin, there will be different in quality and proximate.
Response
The origin and processor of analysed processed fish roes did not showed any influences on results of proximate composition, fatty acid or microbiological analysis, so we consider only the fish species as variability factor. We have no information about the capture area of fish used to extract fish roes, since processed fish roes are a processed product and the origin information is not a label mandatory information. We know only that trout roes came from farmed trout and not from wild animals, and this information has been reflected in the analysis of fatty acids.
The only differences found between manufacturers was the use of additives; for this reason we included this aspect in table 1.
Comment 5
L336: Please revise, it does not read well.
Response
All the discussion about the discriminant analysis has been revised according to the request of reviewer 1
Comment 6
L349: This is the red line for any caviar processor and I dont expect any L. monocytogenes in any caviar product and in EU is 100 cfu/g.
Response
We agree with the reviewer that we expect to not found any L. monocytogenes in fish roes product if they were processed in a well-sanitized laboratory, but from our experiences there are some caviar samples that presented some viable L. monocytogenes (this statement come from analysis performed in our lab, the trial is on going and for this reason data are have not been published). Since the processed fish roes used in present study did not have the chemical characteristic that can classify them in substrate unfavourable to Listeria growth, we perform this analysis, founding all the product comply with the regulatory limit and with the absence of L. monocytogenes.
Comment 7
L352: fish caviar, please do not use roe for processed eggs.
Response
See response to comment 2
Reviewer 3 Report
Please see the notes and annotated Pdf.
Author Response
We haven't any comment for this review as there isn't the file with the reviewer's note.
Round 2
Reviewer 1 Report
Dear Authors,
your manuscript is almost ready for publication, I suggest to the authors 2 recent articles that would improve the quality of their paper for a more modern collocation.
In the Introduction section, you wrote
“Fish roes from other fish species than sturgeon can be defined as caviar substitutes. They are sold at a lower price but they refer to caviar with the aim of exploiting its high appeal among consumers. This reference is evidenced by the habit that some producers have when they confer the black color to products that naturally are differently colored, in order to resemble as much as possible to sturgeon roe [4].”
Authors should consider that caviar substitutes consumption has been investigated for its meaning as a social symbol and that caviar substitutes not necessarily compete with caviar (see Sicuro 2018), please introduce a short comment on this aspect.
Sicuro (2018) The future of caviar production on the light of social changes: a new dawn for caviar? Reviews in Aquaculture 1, 1–16
Multivariate analysis sometimes shows higher explained variance on the first and second component of PCA (see Caredda et al 2018), please introduce a short comment on this aspect.
Caredda M, Addis M, Pes M, Fois N, Sanna G, Piredda G, Sanna G (2018) Physico-chemical, colorimetric, rheological parameters and chemometric discrimination of the origin of Mugil cephalus' roes during the manufacturing process of Bottarga. Food Research International 108 128–135
Finally, I noticed that Figure 1, on loading plot C18:00 is in opposite position respect to C16:00 and SFA along the 2nd axis, that should indicate an inverse proportionality, could they explain this graph?
I included the suggested articles.

Author Response
Dear Authors,
your manuscript is almost ready for publication, I suggest to the authors 2 recent articles that would improve the quality of their paper for a more modern collocation.
Response
The authors are pleased that the changes made to the manuscript have satisfied the reviewer's requests. Here the punctual reply to your final comments.
In the Introduction section, you wrote
“Fish roes from other fish species than sturgeon can be defined as caviar substitutes. They are sold at a lower price but they refer to caviar with the aim of exploiting its high appeal among consumers. This reference is evidenced by the habit that some producers have when they confer the black color to products that naturally are differently colored, in order to resemble as much as possible to sturgeon roe [4].”
Authors should consider that caviar substitutes consumption has been investigated for its meaning as a social symbol and that caviar substitutes not necessarily compete with caviar (see Sicuro 2018), please introduce a short comment on this aspect.
Sicuro (2018) The future of caviar production on the light of social changes: a new dawn for caviar? Reviews in Aquaculture 1, 1–16
Response
A consideration that not all the processed fish roes compete with caviar for the same market, citing the article suggested, has been added.
Multivariate analysis sometimes shows higher explained variance on the first and second component of PCA (see Caredda et al 2018), please introduce a short comment on this aspect.
Caredda M, Addis M, Pes M, Fois N, Sanna G, Piredda G, Sanna G (2018) Physico-chemical, colorimetric, rheological parameters and chemometric discrimination of the origin of Mugil cephalus' roes during the manufacturing process of Bottarga. Food Research International 108 128–135
Response
A comment, citing the reported article, has been added
Finally, I noticed that Figure 1, on loading plot C18:00 is in opposite position respect to C16:00 and SFA along the 2nd axis, that should indicate an inverse proportionality, could they explain this graph?
Response
This consideration is true e punctual. Stearic acid and palmitic acid showed a negative correlation (-0.8). Salmon was the species that presented the highest level of 18:0 and the lowest of 16:0; on the contrary cod, Alaska pollock and capelin were richer of palmitic acid but with the lower stearic acid level. The saturated acid sum is directly linked with 16:0, as this fatty acid in most species represent more than an half of SFA. This aspect placed SFA very close to 16:0 in the PCA plot, but with the slight tendency to move towards 18:0
Reviewer 3 Report
The comments were highlighted using the PDF edit. However, I am satisfied with the reviewer 1 & 2 comments being addressed since they were some repetition.
However, you still need to specify what certified reference material (CRM) did you used in your quantitative analysis for quality control.
Author Response
Response to Reviewer 3 Comments
The comments were highlighted using the PDF edit. However, I am satisfied with the reviewer 1 & 2 comments being addressed since they were some repetition.
Response
Authors are glad that the effort of increasing the quality of our manuscript has been appreciated and recognised by the reviewer
However, you still need to specify what certified reference material (CRM) did you used in your quantitative analysis for quality control.
Response
The provenance of the standards used in our analysis has been added to the text.